# The spreading of SARS-CoV-2: Interage contacts and networks degree distribution

Lucas Sage[1,2‡]*, Marco Albertini[3], Stefani Scherer[1]

**1** Department of Sociology and Social Research, University of Trento, Trento, Italy, **2** Sorbonne Université, GEMASS, Paris, France, **3** Department of Political and social sciences, University of Bologna, Bologna, Italy

‡ Lucas Sage is a joint Trento-Sorbonne doctoral candidate.
* lucas.sage@unitn.it

## Abstract

Notable cross-country differences exist in the diffusion of the Covid-19 and in its lethality. Contact patterns in populations, and in particular intergenerational contacts, have been argued to be responsible for the most vulnerable, the elderly, getting infected more often and thus driving up mortality in some context, like in the southern European one. This paper asks a simple question: is it between whom contacts occur that matters or is it simply how many contacts people have? Due to the high number of confounding factors, it is extremely difficult to empirically assess the impact of single network features separately. This is why we rely on a simulation exercise in which we counterfactually manipulate single aspects of countries' age distribution and network structures. We disentangle the contributions of the kind and of the number of contacts while holding constant the age structure. More precisely, we isolate the respective effects of inter-age contact patterns, degree distribution and clustering on the virus propagation across age groups. We use survey data on face-to-face contacts for Great Britain, Italy, and Germany, to reconstruct networks that mirror empirical contact patterns in these three countries. It turns out that the number of social contacts (degree distribution) largely accounts for the higher infection rates of the elderly in the Italian context, while differences in inter-age contacts patterns are only responsible for minor differences. This suggests that policies specifically targeting inter-age contacts would be little effective.

## Introduction

Early during the Sars-CoV-2 pandemic, scholars from different disciplines rushed to warn the world about the role played by that in-person social contacts have in spreading the virus and in determining the velocity of its diffusion. In line with this warning, many recommended implementing policies—particularly lockdowns—aiming at reducing individuals' social contacts and thus transmissions, and "flatten[ing] the curve" [1]. Demographers and sociologists have further pointed out that specific demographic and behavioral factors–such as a population's age composition, and intergenerational relations or age-mixing of social relations–may be important factors in magnifying the deleterious effects of the SARS-CoV-2 virus, which has far higher lethality levels among older individuals [2–5].

**Data Availability Statement:** The data referred to as POLYMOD data in the article are available at: https://zenodo.org/record/3874557 The data referred to as UN data are available at: https://

population.un.org/wpp/ The data from simulations are within the Supporting Information files.

**Funding:** The authors received no specific funding for this work.

**Competing interests:** The authors have declared that no competing interests exist.

Since age-structure alone cannot account for all cross-country variation in Case Fatality Rates (CFR) [6], we are particularly interested here in those studies that argue that intergenerational relations could explain cross-country differences in the diffusion of the virus among the older population, and thus CFR. The intuition behind these claims is that intergenerational co-residence and contacts led to a faster diffusion of the virus and higher levels of infections in general and particularly among the elderly. This interpretation clearly places a significant importance on the transmission channel from younger to older generations, underlying the vertical dimension of social contacts within families and social groups. This interpretation of the relevance of intergenerational and interage contacts has prompted some scholars to suggest that social distancing measures should specifically focus on limiting interactions between older individuals and the rest of the population [3, 7, 8].

Most articles focusing on the role of intergenerational family relations, however, are rather speculative or provide relatively weak empirical evidence: often the data utilized–e.g. intergenerational co-residence and share of multigenerational households, or overall intergenerational contacts including by phone and mail—are only partially adequate proxies. Further, the analyses are often based on aggregated national-level data and mere correlations with CFR in the early phases of the pandemic [9]. As shown by Arpino et al. [10] the "intergenerational contact hypothesis" (ICH) which can be summarized as follows: *the comparatively higher prevalence of intergenerational co-residence and/or contacts in some countries implies a higher vulnerability to the epidemic that disproportionately affects older adults*, is not convincingly supported by the available data [11–13]. At the same time, the ICH received some partial support from a micro-level study of COVID-19 mortality in the county of Stockholm, based on Swedish register data: the authors show that, among individuals aged 70 or higher, living with someone of working age (i.e. <66 years) was associated with increased COVID-19 mortality risks, together with neighborhood population density and living in a care home [14].

Elsewhere we have suggested that one of the shortcomings of the ICH, and the intuition behind it, is that it focuses exclusively on vertical relations in a society (i.e. between different family generations and/or different age groups), while completely disregarding the important role of horizontal relations, i.e. between same-age or same-generation individuals in a society [15]. In this paper we address one more essential limitation of these studies: when scholars underline the role of national patterns of intergenerational relations—and in general of social relations—in explaining national differences in the spread of the virus among the elderly population, and thus overall CFR, they mix together the effects of different characteristics of networks of social relations: the *degree distribution*, i.e., how many contacts individuals have, and *age-mixing*, i.e., the ages of two or more persons in an interaction, which is also a general proxy of intergenerational contacts. How many contacts a country's citizens have may explain the spreading of the virus better than between whom these contacts take place. An additional important network characteristic is *clustering*, i.e., the tendency for two contacts of an individual to also be in contact, because it can limit the diffusion of viruses [16].

We contribute to the debate generated by the ICH, and the Great Barrington declaration, by carefully disentangling these different dimensions. We analyze the extent to which different national social network patterns could generate cross-country differences in levels of the diffusion of SARS-CoV-2 among the older population. To do this, following Manzo and van de Rijt [17], we use a combination of an agent-based 'susceptible–exposed–infectious–recovered' (SEIR) model of diffusion occurring in networks calibrated on empirical data documenting close-range contacts. We focus on three European countries: Italy, Germany, and Great Britain. Much of the early discussion focused on Italy and its elderly population being particularly hard-hit during the first wave of the pandemic. Germany and Great Britain show sufficiently different contact patterns to make a good comparison.

Our work extends two previous studies. Firstly, [18] showed how country differences in contact frequencies could impact on the effectiveness of exit strategies for lockdowns. Their work shows that cultural habits, that is, how citizens of different countries come into contact with one another, greatly impact the diffusion of the virus. Our analyses tend to confirm that country-specific characteristics of social networks affect the spreading of SARS-CoV-2 in general, and among the older population in particular. We add to [18] by disentangling the "cultural" aspects which make the difference. We find that the number of *contacts* (degree distribution) individuals have, matters much more than the age-mixing patterns. Further analyses confirm that the results hold independently of the underlying age structure and across a large range of clustering differences. Secondly, we build upon the work of [17] that uses similar data to ours to reconstruct a social network respecting the empirical degree distribution. They are interested in the disproportionate role certain individuals with a high number of contacts play in facilitating and accelerating the spread of the virus, and how this could be used by policy makers. Like them, we use realistic networks which respect the asymmetric empirical degree distributions but we also introduce age into the picture.

## Methodological approach

Our strategy consists in 1/ generating a variety of (undirected) social networks with different topological features, 2/ simulating the contagion process over these different networks, and 3/ comparing the diffusion patterns between networks. Doing this, we make sure that the differences observed are attributable only to the specific network properties.

To neutralize the effect of the population's age structure, we keep it constant across simulations: in the results presented in the main text the population of individuals (nodes) is calibrated on the age structure of the Italian population (derived from UN data), and we keep the population size constant (1,500 nodes). In the supplementary material, we rerun all the analysis based on the age structure of Great Britain and of Germany and results are fully confirmed.

To make our social networks resemble country-specific properties of the real world we rely on data from the Polymod survey [19] for three countries: Italy, Germany, and Great Britain. From these data we extract two sets of country-specific information: the age-dependent degree distribution, i.e., how many contacts individuals of different ages have; and the age mix, i.e., the distribution of ages of people with whom individuals of different ages come into contact.

For each network we generate, we select one of the three degree distributions and one of the three age mixes, and create undirected links between the nodes, closely mimicking this empirical information. Importantly, our methodology allows us to generate networks with a degree distribution and an age mix calibrated on different countries.

Overall, this approach allows us to analytically disentangle the role of interage contacts (age mix) from that of the density of social networks (degree distribution), while neutralizing population's age structure. Finally, we also control whether our results hold for different levels of clustering of the networks by fine-tuning clustering levels.

### Empirical data, artificial population networks, and the simulated diffusion process

**Description of empirical networks.**   Polymod data includes a representative sample of individuals living in the countries we considered. Among elements relevant to our analyses, respondents report their daily contacts, their age, and the age of their contacts [19]. Contacts are either physical (defined as "skin-to-skin contact") or non-physical (defined as "a two-way conversation with three or more words in the physical presence of another person"). Participants for whom no information is missing (Italy N = 838; Germany N = 1,286; Great Britain

N = 1,006) were selected to generate information on social networks' interage contacts (age mix) and degree distribution for the three countries considered. S1 Dataset helps visually representing the information contained in the data as ego-centered networks, with the survey's respondents are at the center of the "stars". In Table 1 we report the statistics of these networks for the three countries.

Descriptive statistics, Polymod data, physical and non-physical contacts combined. Because the three surveys covered different numbers of respondents, reporting the gross number of intergenerational ties would be misleading, so we rescaled the values to a comparable, fictional network of 1,500 respondents in each country by multiplying the number of ties found in the data by $1,500/N_{country}$, where $N_{country}$ is the number of respondents in each country survey. The average duration of contacts has been computed using the "middle of the interval" convention (see S1 Appendix), and contacts reported as lasting above four hours are assumed to last six hours.

Age mix is measured by its opposite: age-assortativity $r$, that is the tendency to come into contact with individuals of the same age [20]:

$$r = \frac{\sum_{ij}(A_{ij} - (k_i k_j/2m))x_i x_j}{\sum_{ij}(k_i \delta_{ij} - (k_i k_j/2m)x_i x_j)} \tag{1}$$

where $k_i$ is the degree of node $i$, $\delta_{ij}$ is the Kronecker delta and is set to 1 when $i = j$ and 0 otherwise, $A_{ij}$ is a component in the adjacency matrix (The adjacency matrix A of a network with n nodes is the n×n matrix where the elements $Aij$ equal 1 when there is a link between $i$ and $j$, and 0 when there is no link between them), $m$ is the total number of links in the network, and $x_i$ is the age of node $i$. $r$ is a Pearson correlation coefficient, with a covariance in the numerator and a variance in the denominator. It varies between −1 for a perfectly disassortative network and 1 for a perfectly assortative one, and 0 indicates the absence of any tendency (randomness). Interpreting the values of the assortativity coefficient is therefore straightforward: the higher the coefficient $r$, the more nodes tend to connect with others that are similar in terms of age.

Italy is characterized by a lower age assortativity than Great Britain and Germany, which accords with the view that characterizes Italians as having more intergenerational ties than the other countries. However, these differences seem small. Based on this data, it is not possible to know if this is due to the strength of intergenerational ties. This statistic does not control for the effect of differences in the underlying population's age structure. Thus, the differences could be magnified or diminished if the population of nodes connected by these ties was the same. The network generation method that we adopted allows us to single out the effects of these elements.

The second network property of interest is the degree distribution, i.e., the number of daily contacts (edges). Table 1 reports the mean values, S2 Dataset plots the distributions. In line with the results reported by [17] for France, we find a wide variation within countries in the

**Table 1. Descriptive statistics of contacts declared by nationals of three countries.**

| Country | Italy | Great Britain | Germany |
|---|---|---|---|
| Age assortativity $r$ (lack of age mix) | 0.33 | 0.36 | 0.37 |
| Average degree | 19.74 | 11.74 | 7.97 |
| Total number of contacts in the survey (rescaled to 1,500 individuals) | 29,610 | 17,610 | 11,955 |
| Average duration of contact (minutes) | 144 | 147 | 166 |
| N final sample | 838 | 1,006 | 1,286 |

number of daily contacts between respondents. This skewed distribution is common to many empirical social networks, and is a key feature of complex networks [21]. We preserve this aspect of empirical data in our method of reconstructing networks, to increase the realism of the model. There are large differences in the average number of contacts reported across countries, with Italians reporting many more contacts (about 20 per day) than Germans (about 8 per day) with British in between (about 12). These differences are only partly compensated by Germans having on average longer contacts: see S3 Dataset.

To sum up, the three countries differ empirically to a relevant extent in their network properties, though differences in degree distribution seem to be more pronounced than interage mix. S1 Appendix discusses the empirical data in more depth.

**Artificial population networks.** To assess the extent to which these differences matter for the diffusion of the virus and which age groups are affected most, we built an artificial "national" population to use for the simulation and a counterfactual variation of its characteristics. For each country, first we generated 1,500 nodes with age distribution calibrated on the age distribution of Italy from the UN data. In this data, ages are given in five-year age-bands. We assign each node a notional 'exact' age by allocating the 1,500 first to age-bands (in proportion to the UN data), then evenly within each age-band.

Second, to each node in our artificial population we assigned a degree (number of contacts) corresponding to the empirical degree distribution of a Polymod respondent with the same age. Where several Polymod respondents had the same age as our artificial node, we selected one of these participants at random; if no respondents in the Polymod data had the same age as our artificial node, we searched for the respondent(s) with the age closest to the artificial node and if two respondents had an age equally distant from that of the artificial node, we selected one of the two at random.

Third, in each country, we grouped together all individuals in the Polymod data with a given age (both respondents and declared contacts), and created a unique list of the ages of all persons that came into contact with a person of that age. For instance, suppose a country contains two 20-year-old respondents. We list the ages of all the contacts they declared. Now suppose x other respondents whose ages $\neq 20$ declared having contacts with individuals aged 20. Then the ages of those other respondents are added to the list. In other words, for each tie including a node of age 20, we store the age of the individual at the other end of this tie.

This gives us a distribution of the ages of all contacts of participants of a given age, in a given country. Then we equipped each artificial node with a personal list of contacts' ages, by randomly selecting values from this list $n$ times (with $n$ equal to that node's degree (assigned as explained above)). If there were no Polymod respondents with the same age as the artificial agent, and therefore no list of contacts' ages, we followed the same procedure as for assigning degrees.

After these three steps, all nodes have been assigned a degree—that is, the number of links they are expected to create—and a corresponding list of these contacts' ages. Here, if the node needs to create five links, then the list of contacts will contain five ages, selected as described above.

Fourth, nodes sequentially create links with other available nodes, starting from the node with the highest degree and continuing in descending order. Because they represent a contact in which the two persons are involved, links are always undirected. Every time a node has made the desired number of contacts, it is removed from the pool of available nodes, that is, it can no longer initiate a link, or be selected by other nodes. This part of the algorithm is taken from Manzo and van de Rijt's [17] adaptation of the configuration algorithm (see [22]). Following [17], we forbid multiple links between the same two nodes.

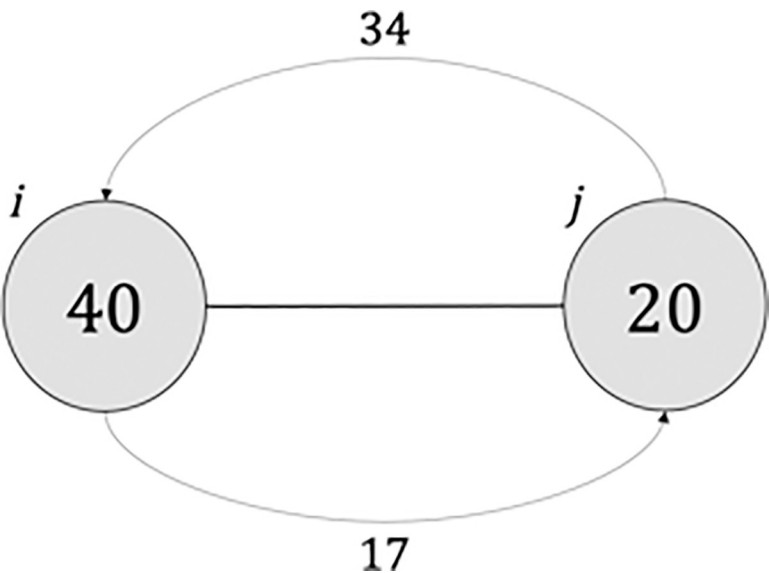

**Fig 1. Example of age-distance computation for an illustrative dyad.**

At this stage, the network matches the empirical degree distribution, but not the empirical age mix. For each link in the network, we defined an "age-distance" $d_{ij}$ between the ideal empirical link and the realized link as:

$$d_{ij} = age_i^{desired} - age_j + age_j^{desired} - age_i \tag{2}$$

Our algorithm aims at minimizing the sum of all links' $L$ age-distances $D$:

$$D = \sum_l^L d_l \tag{3}$$

Fig 1 illustrates the age-distance between ideal and realized links. Node $i$ is 40 years old and the age of the desired contact is 17, while node $j$ is 20 years old and the age of the contact is 34. The age-distance is therefore $|17-20|+|34-40| = 9$. Both nodes connected by a link have an age and a desired age.

The algorithm proceeds in steps by deleting a group of links that is then replaced by the same number of links. We give a detailed description of each step and the criteria of convergence in S2 Appendix. Notice that there is no reason to expect the optimal solution to be $D = 0$. This would imply a perfect match between the structure of ages, and the degree distribution in association with the distribution of desired age. However, for each graph, there is an unknown minimum value of $D$. Given the complexity of the problem, we do not aim at exactly reaching this minimum distance but to get as close to it as possible. To evaluate the proximity between the ideal networks and those realized by our algorithm in Table 2 we report the average distance $D/L$ (total age-distance/total number of links), and the median distance. For all networks, the median age-distance is lower than the mean. This indicates that a few links are relatively distant from the ideal target. However, we see that the medians are relatively small, making us confident that most links are very close to the ideal case. We are therefore confident that, overall, the networks created reflect the age mix contained in the original data.

It is important to note that the first three steps of our algorithm—that is, the calibration against empirical data of (a) the age structure of the population, (b) the distribution of degree and (c) the ages of the contacts matching that degree—are detached from one another. This

**Table 2. Descriptive statistics of simulated networks, based on population age distribution of Italy (UN data).**

|  | Age mixing | Degree distribution | Mean age-distance | Median age-distance | Mean degree | Standard degree | Age assortativity | Clustering | Mean shortest path length |
|---|---|---|---|---|---|---|---|---|---|
| **Empirical** | Germany | Germany | 10.74 | 6.00 | 7.73 | 7.13 | 0.44 | 0.02 | 3.56 |
|  | Great Britain | Great Britain | 16.13 | 7.74 | 10.51 | 7.16 | 0.47 | 0.02 | 3.36 |
|  | Italy | Italy | 13.16 | 5.98 | 16.87 | 11.84 | 0.40 | 0.04 | 2.92 |
| **Counterfactual** | Germany | Great Britain | 11.69 | 5.46 | 10.58 | 7.14 | 0.46 | 0.02 | 3.36 |
|  | Germany | Italy | 10.31 | 5.02 | 16.88 | 11.86 | 0.45 | 0.03 | 2.93 |
|  | Great Britain | Germany | 15.16 | 7.58 | 7.72 | 7.08 | 0.45 | 0.02 | 3.56 |
|  | Great Britain | Italy | 14.82 | 7.14 | 16.81 | 11.79 | 0.44 | 0.03 | 2.91 |
|  | Italy | Germany | 13.99 | 6.78 | 7.71 | 7.06 | 0.39 | 0.02 | 3.57 |
|  | Italy | Great Britain | 15.40 | 6.76 | 10.53 | 7.14 | 0.41 | 0.02 | 3.36 |

allows us to generate counterfactual networks that dissociate these components. By creating networks identical in all properties but one, we can assess the effects of age mix and degree distribution on the diffusion process. For what follows we keep the population age distribution constant and apply three age mixes and three degree distributions, resulting in nine different types of network. S3 Appendix reports the results for each combination with the age structure of each of the three countries considered, resulting in 27 permutations. When the age mix and the degree distribution are aligned on the same country, we call the network "empirical", and when they are not we call it "counterfactual".

Table 2 reports statistics describing the artificial networks. In line with our expectations, the age assortativity is lower in Italy while the average degree is higher. However, the differences in average degree between countries are somewhat smaller than reported in the Polymod data. The average shortest path length is of prime importance for the diffusion processes [23]. It measures the shortest path between a pair of nodes, where a "path" is the lowest number of links necessary to connect the nodes (the geodesic distance). Unsurprisingly, the more links there are in a network—as reflected by the average degree—the lower the average shortest path length. For this reason, we can expect that the virus will spread more easily in networks with a higher average degree. Recall that the population age distribution of the main analysis is the same across all networks (Italy), but we checked the sensitivity of our results to this in S3 Appendix. Lastly, we see that our method generates networks with a very low level of clustering. We come back to this aspect in the last section of the paper.

**Diffusion of the virus: Agent-based SEIR model.** To simulate the diffusion of the virus over the networks, we use a *susceptible–exposed–infectious–recovered* (SEIR) model in which agents unidirectionally move from susceptible $S$, to exposed $E$ (after four days), to infectious $I$ (after four days), to recovered $R$. Each time step $t$ is equal to one day. At each time step, infected individuals can contaminate their susceptible contacts, the probability of which is drawn from a normal distribution $p = 0.05$ $sd = 0.02$. Because there are differences in the average duration of contacts (see S3 Dataset and subsequent comments), we adjusted $p$ for both Germany and Great Britain accordingly—$p_{GER} = p_{IT} * 1.125$ and $p_{GB} = pIT * 1.0208333$—but kept the standard deviation constant. To check the sensitivity of our results to this baseline probability, we varied this value in additional robustness checks but no noteworthy difference was observed (see S6 Appendix). Exposed individuals cannot (yet) contaminate others. An implicit, underlying assumption in previous studies and in our own analyses is that contamination of susceptible individuals is equally probable once they have been in contact with an

infected individual, whatever their age. Thus far, biology and virology studies seem to support this assumption: see for example [24]: Fig 1 and Table 1, or [25].

The simplicity of this SEIR model is helpful when comparing outcomes of various types of network. Only if we wanted to implement various policy scenarios aiming to hamper the propagation of the virus over the network would more complex models—including for example asymptomatic cases, or different behavioral properties between asymptomatic and symptomatic cases—be necessary.

For each of the nine possible networks we generated 50 replications and simulated the diffusion of the virus over each of these networks 50 times. Each time we simulated the spread of the virus, in t = 0 five nodes aged between 26 and 59 years old were chosen at random and assumed to have become infected. We consider a simulation to be over when all nodes are either susceptible or recovered. Overall, for each network type we generated 2,500 data points, each corresponding to the final state of propagation of the virus.

## Results

### Main findings

We start with simple descriptions of the diffusion of the virus. Varying two of the network properties, degree distribution and age mix, according to country-specific patterns, and holding the age structure constant (at that of the Italian population), brings pronounced differences in the simulated diffusion (Fig 2). With the Italian parameters, a larger number of nodes gets infected and this happens at a much faster pace, followed by Great Britain and then Germany.

We now turn to the differences in the age composition of the infected group (holding constant the age structure). Fig 3 displays the stratification of number of cases across age groups

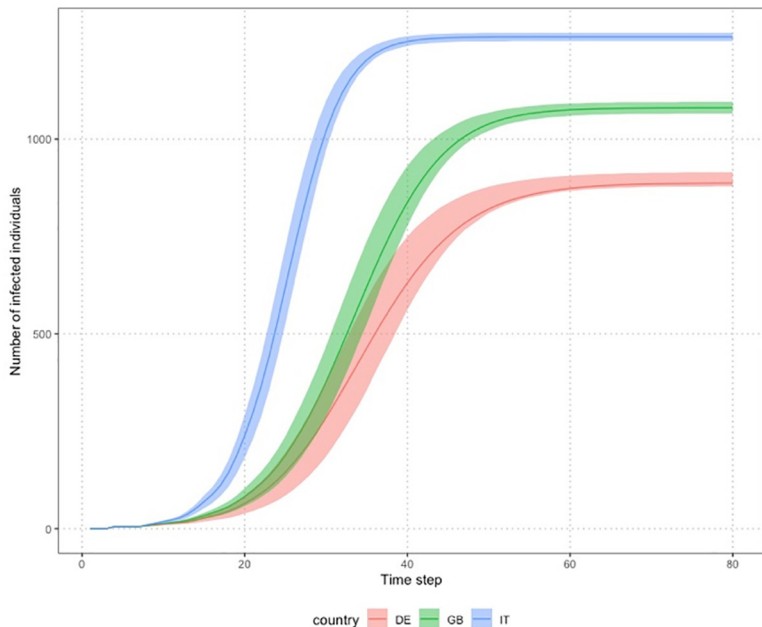

**Fig 2. Average differences in the number of infected individuals over time across countries.** Evolution of the average number of individuals infected over time. Dyadic contagion probability p = 0.05; "empirical" networks of 1,500 nodes, with the Italian population's age structure, and with age mix and degree distribution calibrated on the same country in each panel, but differs across panels. Solid lines represent the average number of cases, the surface around represents the first and third quartile of the distribution.

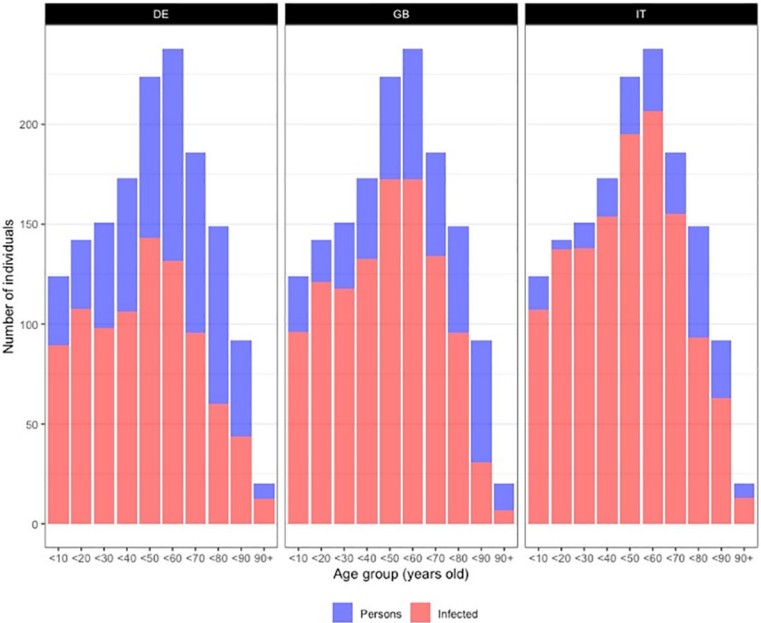

**Fig 3. Countries' average number of individuals infected across age groups.** Italian population's age structure. Dyadic contagion probability p = 0.05. Empirical networks, i.e., degree distribution and age mix, calibrated on the same country in each panel, but differs across panels.

and confirms that national differences in networks do generate large differences in our idealized worlds in all categories. This first result indicates that there exist cultural differences that are significant enough to generate large cross-country differences which is in line with [18]. Importantly, our strategy ensures that these differences are not attributable to differences in the age composition of the population.

However, it is unclear (based solely on these analyses) whether these differences are caused by differences in the age mix or in the degree distribution. Accordingly, in the next step we disentangled the respective contributions of these two factors to the cross-country differences. To do this, we retained the degree distribution calibrated on the Italian data, and applied each country's own age mix. We also tried with the other countries as baseline for the degree distribution, but it did not change the results significantly, either qualitatively or quantitatively. This allowed us to assess the effect of age mix, all other factors (degree distribution and population age structure) held constant. Symmetrically, we retained the age mix calibrated on the Italian data, and applied each country's own degree distribution, to assess the contribution of that element. By comparing the number of cases in networks that are similar in all characteristics but one, we can interpret the resulting differences as being caused by the sole distinguishing characteristic. Figs 4 and 5 plot the impacts of, respectively, the variation in age mix, and of the degree distribution.

The higher age mix of the Italian population (the lower age assortativity) compared to the two other countries seems to cause the older age groups (80+ years) to be slightly more likely to be infected. In other words, if the age structure and number of contacts are identical, the older population tends to be infected more often when the age-pattern of their contacts follows that of the Italian population rather than those of the German or British populations. However, these differences are very small compared to those attributable to the degree distributions displayed in Fig 5. In fact, the higher number of daily contacts reported by the Italians lead to many more infections than the other two countries (who

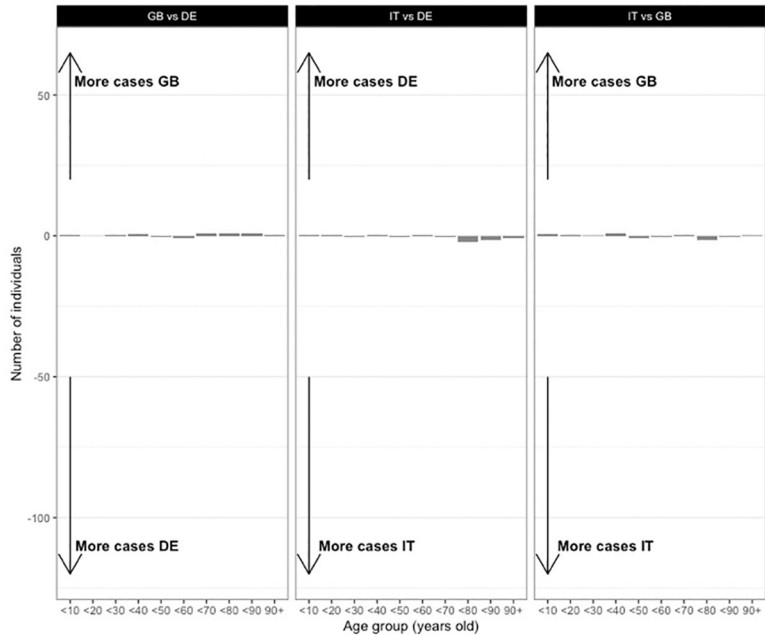

**Fig 4. Effects of age mix on the average number of infected individuals across age groups.** Dyadic contagion probability p = 0.05. Differences in the average number of infected individuals across 10-year age groups at the end of the simulation runs, attributable to differences in age mix *only*. All networks have the population age structure and the degree distribution of Italy. Left panel compares Great Britain with Germany; central panel Germany with Italy; right panel Great Britain with Italy. The x-axis represents equality: both networks have the same number of infected individuals.

report fewer daily contacts), for all age groups, elderly included. As shown in S3 Appendix, these results were confirmed when we ran simulations with varying age distributions. Because our model remains very simple, we do not draw conclusions about the number of cases this could represent in the real world. Our results are however very clear about which of the two network properties matters the most: degree distribution is responsible for large cross-country differences in the number of individuals infected, in the absence of any form of intervention or behavioral change. The patterns of age mix have much less impact on numbers infected. Therefore, it is the higher number of average daily contacts the Italians report that cause more cases in Italy than in Germany or in Great Britain, in almost all age categories. This finding contrasts with parts of the discussion in the media and among academics, who, as outlined in the introduction, have given intergenerational contacts an important role in explaining cross-country differences.

## Assessing the role of clustering

One concern about our methodology is that the networks it generates display low levels of clustering, i.e., contacts between contacts of the respondents. Clustering describes the tendency of networks to form triangles: that is, if node A is linked to B and C, then B and C tend to be linked together (more often than would be expected by chance). Network clustering is measured by the number of closed triplets divided by the number of triplets in the network. Previous research has shown that higher clustering can hamper the propagation of the virus in a network [26] (in the case of COVID-19 see [16]). We checked whether our main result, namely that the average number of contacts in a population matters more than the age mix, is likely to hold at different levels of clustering. We compared Italy with Germany and generated

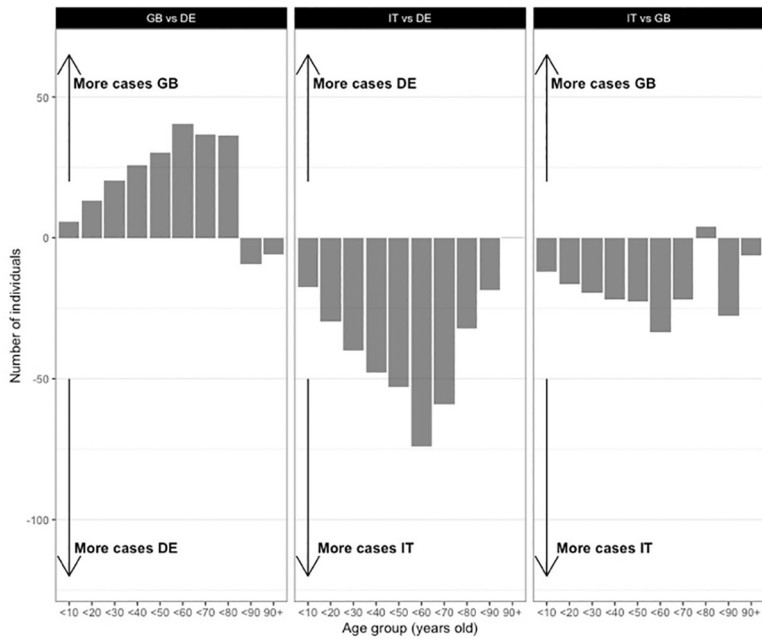

**Fig 5. Effects of degree distribution on the average number of infected individuals across age groups.** Dyadic contagion probability p = 0.05. Differences in the average number of infected individuals across 10-year age groups at the end of the simulation runs, attributable to differences in degree distribution *only*. All networks have the population age structure and the age mix of Italy. Left panel compares Great Britain with Germany; central panel Germany with Italy; right panel Great Britain with Italy. The x-axis represents equality: both networks have the same number of infected individuals.

networks respecting the degree distribution and the age assortativity of the "empirical" networks of the previous section, but with varying levels of clustering. Unfortunately, Polymod data do not provide information about clustering among the participants' contacts, so we modeled a wide range of clustering values.

Two scenarios could challenge our results. First, the effect of differences in the degree distribution could be compensated by higher levels of clustering. Specifically, higher clustering may compensate for the additional infections generated by more contacts. Second, the hampering effect of clustering on the diffusion of the virus could interact in a non-linear way with the effects of age assortativity. For instance, for some level of clustering the difference in age assortativity between the Italian and the German network may greatly slow the propagation of the virus in the Italian network, but have little effect in the German.

To generate clustered networks that respected the empirical degree distribution, we again followed [17] (see S4 Appendix for more details).

Fig 6 left panel plots the differences in the average percentage of elderly people (aged over 65) who become infected, between an Italian network and a German network (see figure legend) for various levels of clustering between 0.05 and 0.35 (which is the maximum clustering value that it is possible to obtain given the German degree distribution). Two results come out of this exercise. First, the diagonal (which represents the difference in the number of cases for networks with similar levels of clustering) shows that the differences found between Italy and Germany in the previous sections are likely to grow bigger for more clustered networks. This means that clustering hampers the diffusion of the virus to a greater extent in the German network than in the Italian one (see also S5 Appendix). This is due precisely to the difference in the degree distribution between the two networks, with the Italians having on average more

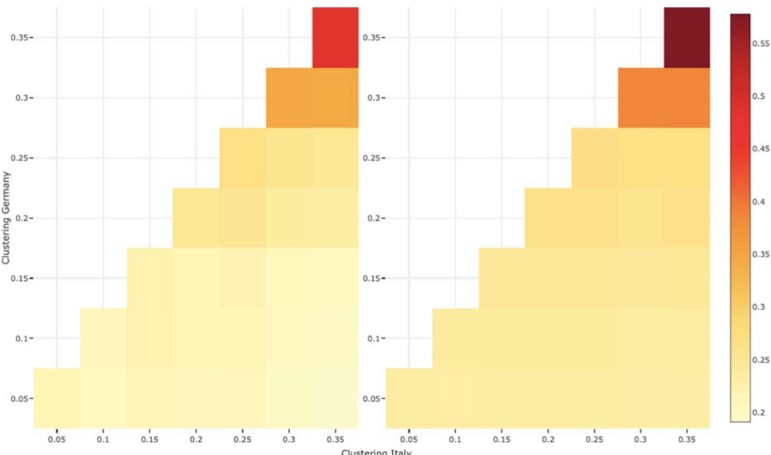

**Fig 6. Effect of clustering on the difference between Italy and Germany.** Average differences in the proportion of cases between an Italian network (age assortativity = 0.40 and degree distribution of Italy) and a German network (age assortativity = 0.44 and degree distribution of Germany), for 1,250 replications for each cell (25 networks * 50 simulations of the diffusion of the virus, starting from different originally infected nodes), for different values of clustering. Dyadic contagion probability $p = 0.05$. Left panel represents the difference in the proportion of cases among the elderly (nodes aged over 65); right panel: for the full population.

daily contacts than the Germans. The second result of interest is that, as the Italian network gets increasingly more clustered than the German network, we see the differences between the number of cases among the elderly reduce slightly (this also happens in younger age groups). However, even when the clustering difference reaches the maximum possible value, there are still about 20% more cases in Italy than in Germany. This means that our principal result holds: *in silico* differences in the network, and more precisely differences in the degree distribution, make major differences to the spread of the disease.

These results are confirmed by a full exploration of the parameter space around clustering and age assortativity, which results are reported in S5 Appendix. We investigated whether age assortativity plays any role in the diffusion patterns, and whether its effect could depend on the degree distribution. Our results indicate the effect of clustering largely depends on the degree distribution, which is in line with theoretical results [27]: the same amount of clustering generates a larger reduction in the number of cases in Germany than in Italy. Moreover, clustering reduces the spread of the disease much more than age assortativity. Nonetheless, we find that age assortativity has a hampering effect for all values of clustering, and so it could reduce the number of cases among the elderly. But for this effect to appear would require differences in age assortativity that are far too distant from the empirical cross-country differences documented in Table 2 to be likely in realistic settings. This explains why we did not find any effect of age assortativity in Fig 4. Results are similar when we considered the timing of the contagion peak as the outcome of interest (see S5 Appendix).

## Discussion

Previous research has found a range of social aspects to be at the root of cross-country differences in the spread of the COVID-19 pandemic. In line with previous research [17, 27], our results confirm that, all else being constant, social networks of contacts potentially play a major role in the diffusion of the pandemic. Based on an agent-based SEIR model simulating the spread of the virus over social networks in three different countries, we showed how the number and kind of social contacts can indeed contribute to explaining country differences in

the diffusion among different age groups. In contrast to previous discussions, our results show that cross-country differences in the number of infected cases are more attributable to how many contacts individuals have than to the age of the persons they meet. Our simulations revealed that differences in the empirical age mix of contacts in European countries do not lead to sharp differences in the number of infected individuals. Rather than supporting the hypothesis that intergenerational (and, more generally, interage) contacts are at the root of a fast and important diffusion among the older population, our analyses pointed to a second network property that has a greater extent on diffusion of the virus: degree distribution, or the sociability of members of a society. We found remarkable differences between countries that led *in silico* to large differences in the number of cases ultimately found in our simulations, in particular among the elderly. The very limited effect of age mix is unrelated to the age structure of the population. We also checked whether the extra cases attributable to the higher sociability that Italians report (compared to Germans) could be compensated by Italians also "clustering" more, but we did not find anything to call our main result into question. Clustering hampers the diffusion of the virus more in Germany than it does in Italy. Even when we increased clustering in Italy but not in Germany, the differences in the number of cases only diminished slightly.

We want to underline that our model is not designed as a basis for making policy recommendations. Instead, we explicitly aimed to create an artificial world in which we could control all factors, including the demographic structure of populations. Accordingly, we may have overestimated the contribution of social networks to the cross-country differences. We also did not build any form of behavioral change into any stage of the diffusion process. Our purpose was precisely to leave these aspects aside, something which would not have been possible with empirical data, where lockdowns and other policy strategies come into play and may interact in complex ways with the underlying network structures. We believe that this is both a strength and a weakness in this study and the methodology that it developed.

That said, some indications do emerge. Firstly, our analyses could be interpreted as giving precision to vague notions such as between-country cultural differences. We find that there do, indeed, exist cultural differences in the way citizens of different countries come into contact with each other on a daily basis. More importantly, we find that these differences impact on the spread of the disease. This calls for care in evaluating national non-pharmaceutical policy interventions. Such evaluations necessarily rely on the constitution of a control group that is supposed to constitute the counterfactual, i.e., "what would have happened had the intervention not occurred". Our analyses, like [18], suggest that countries might not react in the same way to these interventions, because of different underlying network structures.

Secondly, all else being constant, policy interventions specifically targeting intergenerational relations within families to protect the elderly may produce small results. More generally, interventions that aim at reducing how many contacts a population has with one another, or even targeting individuals with high number of contacts [17] could have greater or less effect in different countries. Countries with denser social networks may have to implement stricter limitations in order to obtain results similar to their neighbors'.

## Supporting information

**S1 Appendix. Polymod data analyses.**
(DOCX)

**S2 Appendix. Description of the algorithm to reduce total age-distance D.**
(DOCX)

**S3 Appendix. Varying population age structure.**
(DOCX)

**S4 Appendix. Generation of networks with tunable levels of clustering and age assortativity.**
(DOCX)

**S5 Appendix. Further examination of the roles of age assortativity and clustering.**
(DOCX)

**S6 Appendix. Alternative probability of dyadic contagion.**
(DOCX)

**S1 Dataset. dataMainItaly.**
(CSV)

**S2 Dataset. dataMainGermany.**
(CSV)

**S3 Dataset. dataMainGreatBritain.**
(CSV)

**S4 Dataset. dataFig6.**
(CSV)

**S5 Dataset. dataAppendix5.**
(CSV)

## Author Contributions

**Conceptualization:** Lucas Sage, Marco Albertini, Stefani Scherer.

**Data curation:** Lucas Sage.

**Formal analysis:** Lucas Sage.

**Funding acquisition:** Stefani Scherer.

**Methodology:** Lucas Sage.

**Software:** Lucas Sage.

**Supervision:** Marco Albertini, Stefani Scherer.

**Visualization:** Lucas Sage.

**Writing – original draft:** Lucas Sage, Marco Albertini.

**Writing – review & editing:** Lucas Sage, Marco Albertini, Stefani Scherer.

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
