## [Decision Letter · Decision Letter 0]

18 May 2021

PONE-D-21-09973

Dissecting the roles of inter-age contacts and networks degree distribution in the spreading of SARS-CoV-2. An Agent-Based Model using empirical network data.

PLOS ONE

Dear Dr. Sage,

Thank you for submitting your manuscript to PLOS ONE. After careful consideration, we feel that it has merit but does not fully meet PLOS ONE’s publication criteria as it currently stands. Therefore, we invite you to submit a revised version of the manuscript that addresses the points raised during the review process.

The most important point to address in the revision, that has been pointed out by both the reviewers, is the analysis of the relationship between age distributions and age assortativity patterns. 

We look forward to receiving your revised manuscript.

Kind regards,

Floriana Gargiulo

Academic Editor

PLOS ONE

Journal Requirements:

Reviewers' comments:

Reviewer's Responses to Questions

**Comments to the Author**

1. Is the manuscript technically sound, and do the data support the conclusions?

Reviewer #1: No

Reviewer #2: Yes

2. Has the statistical analysis been performed appropriately and rigorously? 

Reviewer #1: Yes

Reviewer #2: Yes

3. Have the authors made all data underlying the findings in their manuscript fully available?

Reviewer #1: Yes

Reviewer #2: Yes

4. Is the manuscript presented in an intelligible fashion and written in standard English?

Reviewer #1: Yes

Reviewer #2: Yes

5. Review Comments to the Author

Reviewer #1: The authors study the impact of different face-to-face meeting social networks on the diffusion of the virus SARS-COV-2 in a population. They use an agent-based model to experiment, especially to control the properties of the network regarding the average number of meetings by agent, and the level of contacts between agents having different classes of ages. They aim to know what has the greatest impact on the diffusion of the virus: the inter-age contact or the average number of relationships a day. They built their network based on surveyed meeting data in Germany, England and Italy. They show that the average number of relationships by day is far more important for the level of diffusion compared to the level of inter-age contact.

The paper is clear, well written and deserves to be published if improved. The research question is clear and pretty well argued. I have only one concern with their methodology which assumes that the observed age-mix in real data is independent from the age structure. From this assumption, they argue to control the impact of the different properties in their experimental design using only the age distribution of Italy.

However this assumption can be false since we assume that depending on the local distribution of ages (especially when a class of age is over or under represented), an individual can be constrained regarding the average number of contacts, or the number of contacts with a given class of ages. Thus, their methodology should include age distribution in the experimental design, considering 27 cases with 3 age distribution, 3 age-mix and 3 average degrees, to conclude about the effect of average degree versus age-mix. This is only by experimenting and presenting the results from these compared 27 cases that they can robustly conclude about the network’s property implying the strongest impact on the diffusion of the virus. This should not be a great deal for the authors who say that they have tested their results with other age distributions without presenting their results.

Detailed comments:

Page 4, end of the second paragraph: you wrote “We finally also control if your results hold for….”, I guess you want to wrote “if our results”.

Page 4 and 5, end of the pages: …………………….

Page 5, table 1: please precise the type of contact you talked about (physical, ….)

Page 8, 4th line, what is the distance D/L ?

Page 8, second paragraph. The authors make the assumption that the age-mix in a population is independent from its age distribution. Then their methodology argues this is sufficient to vary age-mix meetings and average number of contacts for an arbitrary chosen age distribution to conclude about the impact of age-mix and average degree. However, if we consider age-mix and age distribution can be dependent from age distribution, the methodology should include age distribution in the experimental design, considering 27 cases with 3 age distribution, 3 age-mix and 3 average degrees, to conclude about the effect of average degree versus age-mix.

Page 10, how many times last the simulation? Do you compare the diffusion for a same horizon for all the simulated social networks, or by the end of the diffusion process?

Page 12, comments on figure 4. The differences of results between the different cases seem to be very small, did the authors check that they are statistically significant?

Appendix S2, page 7: in the middle of the page, there is a debate to precise around physical or not contacts.

Appendix S5, page 14, last paragraph : “in on a network….”

Appendix S5, page 15, last paragraph, what is ICH?

Appendix S5, page 16: the sentence “if even in such conditions age assortativity as so little effect, it is unlikely that we could detect something in the real work were others factors come into play” is really not convincing since you can have some strong interactions effects of age assortativity only in the presence of other factors!

Reviewer #2: Review of manuscript PONE-D-21-09973 : « Dissecting the roles of inter-age contacts and networks degree distribution in the spreading of SARS-CoV-2. An Agent-based Model using empirical network data »

The paper looks at how inter-country variations in the spread of the SARS-CoV-2 virus can originate in different patterns of social interactions. Its main originality lies in distinguishing between the frequency of inter-individual contacts and the age-structure of these contacts. It shows that the former dimension is much more important than the latter in explaining inter-country variations in contamination rates. This is true even when due account is taken of differences in clustering. The methodological approach consists of comparing three European countries, Italy, Germany and the United Kingdom, using Italy as a benchmark to control for all possible disturbing factors, the population’s age structure in particular. Simulations are then carried out by using the social interaction characteristics specific to each country.

This is well-executed paper that makes a well-focused argument based on a sound methodology that the authors obviously master. Moreover, it makes an interesting contribution by offering a conclusion that is not a priori evident and carries some important policy implications. For these reasons, the paper deserves to be published. I have nevertheless a few remarks and suggestions that I believe should be taken into account before the paper is definitely accepted for publication. The revision should not take much time as there is no new simulation work required.

1. The authors ought to cite previous works that deal with the same issue as their own, and do so in some detail. In particular, the reader should be told more about the results obtained by Manzo and de Rijt [17]. Also, there is no mention of a paper that is directly relevant to the effort of the present authors : J.P. Platteau and V. Verardi, 2020. “How to Exit Covid-19 Lockdowns: Culture Matters”, Covid Economics: Vetted and Real-Time Papers, CEPR (Centre for Economic Policy Research), Issue 23 (May). Platteau and Verardi also follow a simulation approach based on the SEIR model, and they apply country-specific interactions matrices to a reference country in order to study the impact of social interactions on the spread of the virus. Moreover, their work is based on a triad composed of Italy, Germany, and Belgium (the benchmark country), with Belgium playing the role of the intermediate country as well (like the UK in the present case). As for the differences, although they take account of the role of age in determining interactions, they do not separate the effects of interaction frequencies and age mixing. On the other hand, they study the impact of interaction frequencies on different lockdown strategies.

2. As it is written, the authors assume that the reader is familiar with graph theory. I think they could do better in explaining their method. For example, on page 5, they speak about an “adjency matrix”, a misnomer for what is actually called an “adjacency matrix” in graph theory. They should definitely explain a bit what it consists of. Moreover, equation (1) measuring age assortativity should be explained. As written, it is hard to understand why it captures age assortativity. Also, since Aij is a matrix and not a number, it cannot appear under a summation sign as such (see both the numerator and the denominator of eq. (1)). I suppose that the authors mean the component aij of the matrix. If yes, it should be written thus. Finally, on page 7, second sentence, what does “with length equal to the desired degree” mean precisely in this context?

3. The authors implicitly assume that the probability of contaminating or being contaminated is the same regardless of the ages of the persons in contact. I think they should provide some justification for this assumption.

4. The title is much too long and could be simplified without losing content. In addition, the subtitle “Analytical approach” (page 4) is better replaced by “Methodological approach”.

6. PLOS authors have the option to publish the peer review history of their article (what does this mean?). If published, this will include your full peer review and any attached files.

Reviewer #1: No

Reviewer #2: **Yes: **Jean-Philippe Platteau, Active Emeritus Professor in Economics, University of Namur.

---

## [Author Response · Author response to Decision Letter 0]

2 Jul 2021

Dear Editor and Reviewers, 

first and foremost, we would like to thank the editor and the two reviewers for the careful reading, the attention that has been dedicated to our work and the valuable comments and suggestions we got. We believe that they helped us strengthening the manuscript. With regards to the most important point raised by the reviews, we performed and report the requested additional analyses (we place them in the supplementary material as the substantive results do not differ from the ones reported in the article); we included the requested specifications, additions and clarifications; and we had the text proof read. The copyediting of the manuscript for language usage, spelling, and grammar has been provided by: John Firth Proofreader & Copy Editor (john-firth-editor.co.uk) see also: https://www.ciep.uk/directory/john-firth.

Below we respond in detail to the comments of the editor and of the two reviewers. 

As requested, we submit two versions of the revised manuscript, a clean one and one with track changes. As you will realize, side by the detailed comments of the reviewers, we also dedicated quite a bit of attention to the language. 

We hope the changes to the manuscript correspond to the expectations and that the revised version can be considered for publication. 

Journal Requirements:

Authors’ response:

We carefully checked the new list of references. In addition to the previous references, we added: 

J.P. Platteau and V. Verardi, 2020. “How to Exit Covid-19 Lockdowns: Culture Matters”, Covid Economics: Vetted and Real-Time Papers, CEPR (Centre for Economic Policy Research), Issue 23 (May) 

T.C. Jones, G. Biele G, B. Mühlemann, T. Veith, J. Schneider, J. Beheim-Schwarzbach, T. Bleicker, J. Tesch, M.e L. Schmidt, L. E. Sander, F. Kurth, P. Menzel, R. Schwarzer, M. Zuchowski, J. Hofmann, A. Krumbholz, A. Stein, A. Edelmann, V. M. Corman, and Ch. Drosten (2021) Estimating infectiousness throughout SARS-CoV-2 infection course, Science, 10.1126/science.abi5273 (2021). 

Baggio S, L'Huillier AG, Yerly S, Bellon M, Wagner N, Rohr M, Huttner A, Blanchard-Rohner G, Loevy N, Kaiser L, Jacquerioz F, Eckerle I. SARS-CoV-2 viral load in the upper respiratory tract of children and adults with early acute COVID-19. Clin Infect Dis. 2020 Aug 6:ciaa1157. doi: 10.1093/cid/ciaa1157. Epub ahead of print. PMID: 32761228; PMCID: PMC7454380.

Authors’ response:

Copyediting of the manuscript for language usage, spelling, and grammar has been provided by: John Firth Proofreader & Copy Editor (john-firth-editor.co.uk) see also: https://www.ciep.uk/directory/john-firth

Reviewer #1: The authors study the impact of different face-to-face meeting social networks on the diffusion of the virus SARS-COV-2 in a population. They use an agent-based model to experiment, especially to control the properties of the network regarding the average number of meetings by agent, and the level of contacts between agents having different classes of ages. They aim to know what has the greatest impact on the diffusion of the virus: the inter-age contact or the average number of relationships a day. They built their network based on surveyed meeting data in Germany, England and Italy. They show that the average number of relationships by day is far more important for the level of diffusion compared to the level of inter-age contact.  The paper is clear, well written and deserves to be published if improved. The research question is clear and pretty well argued. I have only one concern with their methodology which assumes that the observed age-mix in real data is independent from the age structure. From this assumption, they argue to control the impact of the different properties in their experimental design using only the age distribution of Italy.  However this assumption can be false since we assume that depending on the local distribution of ages (especially when a class of age is over or under represented), an individual can be constrained regarding the average number of contacts, or the number of contacts with a given class of ages. Thus, their methodology should include age distribution in the experimental design, considering 27 cases with 3 age distribution, 3 age-mix and 3 average degrees, to conclude about the effect of average degree versus age-mix. This is only by experimenting and presenting the results from these compared 27 cases that they can robustly conclude about the network’s property implying the strongest impact on the diffusion of the virus. This should not be a great deal for the authors who say that they have tested their results with other age distributions without presenting their results. 

Authors’ response: 

Reviewer 1 questioned the validity of an underlying assumption behind our results: the independence between age distribution and age mix. In our analysis the age distribution was held constant and calibrated on Italy. Reviewer 1 raised the concern that age distribution and age mixing are not independent and that we might confound the two by not varying them independently. We thank reviewer 1 for pointing this out, and we now report in supplementary material Appendix S3 the new additional results. As reviewer 1 underlines, 27 unique combinations are possible: 3 age distributions, 3 age-mixes and 3 degree distributions. Presenting these results in a concise way was challenging, because our analyses relies on comparison between pairs of networks. We opted for a solution where we reproduced our two principal figures, namely Fig 4 and 5. Fig S5_1 reproduces Fig 4 that aimed at understanding the effect of age mix on virus’ spread holding constant the age distribution and the degree distribution. In the nine images composing Fig S5_1 we used all combinations of underlying age distribution + degree distribution (3*3=9). Symmetrically in Fig S5_2 we reproduce Fig 5 with all combinations of underlying age distribution and age mix (9). Results show no major deviation from the baseline scenario. We interpret this as a confirmation of our original results which is why we place these additional analyses in the supplementary material and report in the main text only the variation of age-mix and degree distribution as in the initial version. 

 Detailed comments: Page 4, end of the second paragraph: you wrote “We finally also control if your results hold for….”, I guess you want to wrote “if our results”.

Authors’ response: 

We fixed that issue.

 Page 4 and 5, end of the pages: …………………….

Authors’ response: 

We fixed that typo.

 Page 5, table 1: please precise the type of contact you talked about (physical, ….)

Authors’ response: 

We now mention in Table 1’s legend that the statistics describe all types of contacts together.

 Page 8, 4th line, what is the distance D/L ?

Authors’ response: 

D/L was the total age distance (the sum of all links’ age distance) divided by the number of links, i.e. the average age distance. We added a short description in parentheses right after that comment. 

 Page 8, second paragraph. The authors make the assumption that the age-mix in a population is independent from its age distribution. Then their methodology argues this is sufficient to vary age-mix meetings and average number of contacts for an arbitrary chosen age distribution to conclude about the impact of age-mix and average degree. However, if we consider age-mix and age distribution can be dependent from age distribution, the methodology should include age distribution in the experimental design, considering 27 cases with 3 age distribution, 3 age-mix and 3 average degrees, to conclude about the effect of average degree versus age-mix.

Authors’ response: 

See our response to the first comment of reviewer, in which we detail how we addressed this issue. 

 Page 10, how many times last the simulation? Do you compare the diffusion for a same horizon for all the simulated social networks, or by the end of the diffusion process?

Authors’ response: 

To clarify this point we added a short sentence in page 12 at the end of the section where we describe the SEIR agent based model contagion process: 

“We consider a simulation to be over when all nodes are either susceptible or recovered.”

 Page 12, comments on figure 4. The differences of results between the different cases seem to be very small, did the authors check that they are statistically significant?

Authors’ response: 

We did not perform statistical tests for two reasons. Firstly, when dealing with simulated data, we believe that statistical tests can be misleading. Although it is a debated question, we believe that any difference could become statistically significant if we would perform more simulations, thereby increasing the sample size and the power of the test. Secondly, in figure 4, we were more interested in the overall patterns and tendencies rather than in any specific point of the parameter space. In other words, what figure 4 shows is that 1/ even if clustering was higher in Italy than in Germany the number of cases would still be largely higher in Italy (at least than 20% cases more, as shown by the lower limit of the color scale) 2/ for the same level of clustering in the two countries, the higher the clustering the more the differences between Germany and Italy increase (the diagonal). For instance, knowing whether increasing clustering in Italy from 0.30 to 0.35, all else being constant, is statistically significant, is not the kind of differences that we comment on, and it would not add much to the substantive contribution of figure 4. 

 Appendix S2, page 7: in the middle of the page, there is a debate to precise around physical or not contacts.

Authors’ response: 

We now believe that the distinction between physical and non-physical contacts is clearer. In our main analyses, we use both indistinctively. 

 Appendix S5, page 14, last paragraph : “in on a network….”

Authors’ response: 

We fixed that issue. 

 Appendix S5, page 15, last paragraph, what is ICH?

Authors’ response: 

ICH refer the intergenerational contact hypothesis. We now use the abbreviation in the main text only, and explicitly mention “intergenerational contact hypothesis” in the appendix to make it clearer.

  Appendix S5, page 16: the sentence “if even in such conditions age assortativity as so little effect, it is unlikely that we could detect something in the real work were others factors come into play” is really not convincing since you can have some strong interactions effects of age assortativity only in the presence of other factors!

Authors’ response: 

We thank reviewer 1 for this comment. We modify this part of the appendix in the new version in the following way (page 16 supplementary material):

“It is worth recalling that our artificial worlds only consider three factors potentially impacting on the virus diffusion processes: degree distribution, age mixing, and population age structure. How each of these properties may interact with other factors in the real world could increase or decrease their respective importance. Further work, progressively incorporating other relevant factors, is needed to understand whether those factors could interact with age-mixing differences and thereby increase the effects of age mixing on the diffusion of the virus across a network.”

Reviewer #2: Review of manuscript PONE-D-21-09973 : « Dissecting the roles of inter-age contacts and networks degree distribution in the spreading of SARS-CoV-2. An Agent-based Model using empirical network data »  The paper looks at how inter-country variations in the spread of the SARS-CoV-2 virus can originate in different patterns of social interactions. Its main originality lies in distinguishing between the frequency of inter-individual contacts and the age-structure of these contacts. It shows that the former dimension is much more important than the latter in explaining inter-country variations in contamination rates. This is true even when due account is taken of differences in clustering. The methodological approach consists of comparing three European countries, Italy, Germany and the United Kingdom, using Italy as a benchmark to control for all possible disturbing factors, the population’s age structure in particular. Simulations are then carried out by using the social interaction characteristics specific to each country. This is well-executed paper that makes a well-focused argument based on a sound methodology that the authors obviously master. Moreover, it makes an interesting contribution by offering a conclusion that is not a priori evident and carries some important policy implications. For these reasons, the paper deserves to be published. I have nevertheless a few remarks and suggestions that I believe should be taken into account before the paper is definitely accepted for publication. The revision should not take much time as there is no new simulation work required.

 1. The authors ought to cite previous works that deal with the same issue as their own, and do so in some detail. In particular, the reader should be told more about the results obtained by Manzo and de Rijt [17]. Also, there is no mention of a paper that is directly relevant to the effort of the present authors : J.P. Platteau and V. Verardi, 2020. “How to Exit Covid-19 Lockdowns: Culture Matters”, Covid Economics: Vetted and Real-Time Papers, CEPR (Centre for Economic Policy Research), Issue 23 (May). Platteau and Verardi also follow a simulation approach based on the SEIR model, and they apply country-specific interactions matrices to a reference country in order to study the impact of social interactions on the spread of the virus. Moreover, their work is based on a triad composed of Italy, Germany, and Belgium (the benchmark country), with Belgium playing the role of the intermediate country as well (like the UK in the present case). As for the differences, although they take account of the role of age in determining interactions, they do not separate the effects of interaction frequencies and age mixing. On the other hand, they study the impact of interaction frequencies on different lockdown strategies.

Authors’ response: 

We agree with R2 that in the previous version of the manuscript we missed a relevant piece of previous research and apologize for this lack of attention. We follow the reviewer suggestion and we discuss the directly relevant literature in the text now and we better specify the contribution of our paper vis-à-vis previous papers adopting a similar approach (including those already cited in the previous version): 

J.P. Platteau and V. Verardi, 2020. “How to Exit Covid-19 Lockdowns: Culture Matters”, Covid Economics: Vetted and Real-Time Papers, CEPR (Centre for Economic Policy Research), Issue 23 (May) 

 2. As it is written, the authors assume that the reader is familiar with graph theory. I think they could do better in explaining their method. For example, on page 5, they speak about an “adjency matrix”, a misnomer for what is actually called an “adjacency matrix” in graph theory. They should definitely explain a bit what it consists of. Moreover, equation (1) measuring age assortativity should be explained. As written, it is hard to understand why it captures age assortativity. Also, since Aij is a matrix and not a number, it cannot appear under a summation sign as such (see both the numerator and the denominator of eq. (1)). I suppose that the authors mean the component aij of the matrix. If yes, it should be written thus. Finally, on page 7, second sentence, what does “with length equal to the desired degree” mean precisely in this context?

Authors’ response: 

We thank reviewer 2 for these comments. First of all, we corrected “adjency matrix” for “adjacency matrix”, and added a footnote (footnote 4) explaining what it consists in. Secondly, we modified the formulae of equation (1) going back to the original formulation of Newman 2018’s book “Networks” that we had slightly modified in the first version of the manuscript. We believe that it will be easier for readers who would be interested in reading more about the assortativity coefficient in the original book, to have a correspondence between our text and the original book. Compared to the previous version we introduced the Kronecker delta (which we explain in the text). We also added sentences in the text (on page 6) helping the reader to understand the logic behind the assortativity coefficient. Moreover, we thank reviewer 2 for pointing out that Aij is an element of the matrix and not the matrix itself. We corrected this mistake in the text describing equation (1). Finally, we deleted the sentence “with length equal to the desired degree”, because it created confusion in the reader’s mind. We re-wrote this section in a clearer way. Specifically, this sentence has been replaced by the following one on page 7:

“Then we equipped each artificial node with a personal list of contacts’ ages (for simulation purposes), by randomly selecting values from this list n times (with n equal to that node’s degree (assigned as explained above)).”

 3. The authors implicitly assume that the probability of contaminating or being contaminated is the same regardless of the ages of the persons in contact. I think they should provide some justification for this assumption.

Authors’ response: 

We now include references to back up our assumption that the probability of contamination does not vary with age: 

T.C. Jones, G. Biele G, B. Mühlemann, T. Veith, J. Schneider, J. Beheim-Schwarzbach, T. Bleicker, J. Tesch, M.e L. Schmidt, L. E. Sander, F. Kurth, P. Menzel, R. Schwarzer, M. Zuchowski, J. Hofmann, A. Krumbholz, A. Stein, A. Edelmann, V. M. Corman, and Ch. Drosten (2021) Estimating infectiousness throughout SARS-CoV-2 infection course, Science, 10.1126/science.abi5273 (2021). 

Baggio S, L'Huillier AG, Yerly S, Bellon M, Wagner N, Rohr M, Huttner A, Blanchard-Rohner G, Loevy N, Kaiser L, Jacquerioz F, Eckerle I. SARS-CoV-2 viral load in the upper respiratory tract of children and adults with early acute COVID-19. Clin Infect Dis. 2020 Aug 6:ciaa1157. doi: 10.1093/cid/ciaa1157. Epub ahead of print. PMID: 32761228; PMCID: PMC7454380.

 4. The title is much too long and could be simplified without losing content. In addition, the subtitle “Analytical approach” (page 4) is better replaced by “Methodological approach”.

Authors’ response: 

We modified and shortened the title of the article, it now reads: “The spreading of SARS-CoV-2: inter-age contacts and networks degree distribution”. We also changed the section’s name to “methodological approach”.

---

## [Editor Report · Decision Letter 1]

29 Jul 2021

The spreading of SARS-CoV-2:  interage contacts and networks degree distribution

PONE-D-21-09973R1

Dear Dr. Sage,

We’re pleased to inform you that your manuscript has been judged scientifically suitable for publication and will be formally accepted for publication once it meets all outstanding technical requirements.

Kind regards,

Floriana Gargiulo

Academic Editor

PLOS ONE
---

## [Editor Report · Acceptance letter]

11 Aug 2021

PONE-D-21-09973R1 

The spreading of SARS-CoV-2: interage contacts and networks degree distribution 

Dear Dr. Sage:

I'm pleased to inform you that your manuscript has been deemed suitable for publication in PLOS ONE. Congratulations! Your manuscript is now with our production department. 

Kind regards, 

on behalf of

Dr. Floriana Gargiulo 

Academic Editor

PLOS ONE